

# 1    Disappearing day-of-week ozone patterns in US nonattainment areas

Heather Simon[1], Christian Hogrefe[2], Andrew Whitehill[2], Kristen M. Foley[2], Jennifer Liljegren[3],
Norm Possiel[1], Benjamin Wells[1], Barron H. Henderson[1], Lukas C. Valin[2], Gail Tonnesen[4], K.
Wyat Appel[2], Shannon Koplitz[1]
[1]US Environmental Protection Agency, Office of Air and Radiation, Research Triangle Park, NC
[2]US Environmental Protection Agency, Office of Research and Development, Research Triangle Park, NC
[3]US Environmental Protection Agency, Region 5, Chicago, IL
[4]US Environmental Protection Agency, Region 8, Denver, CO
*Correspondence to*: Heather Simon (Simon.Heather@epa.gov)
**Abstract.** Past work has shown that traffic patterns in the US and resulting $NO_X$ emissions vary by day of week, with
$NO_X$ emissions typically higher on weekdays than weekends. This pattern of emissions leads to different levels of
ozone on weekends versus weekdays and can be leveraged to understand how local ozone formation changes in
response to $NO_X$ emissions perturbations in different urban areas. Specifically, areas with lower $NO_X$ but higher ozone
on the weekends (the weekend effect) can be characterized as $NO_X$ -saturated and areas with both lower $NO_X$ and
ozone on weekends (the weekday effect) can be characterized as $NO_X$-limited. In this analysis we assess ozone
weekend-weekday differences across US nonattainment areas using 18 years of observed and modeled data from
2002-2019 using two metrics: mean ozone and percentage of days > 70 ppb. In addition, we quantify the modeled and
observed trends in these weekend-weekday differences across this period of substantial $NO_X$ emissions reductions in
the US. The model assessment is carried out using EPA's Air QUAlity TimE Series Project (EQUATES) CMAQ
dataset. We identify 3 types of ozone trends occuring across the US: disappearing weekend effect, disappearing
weekday effect, and no trend. The disappearing weekend effect occurs in a subset of large urban areas that were $NO_X$
-saturated (i.e., VOC-limited) at the beginning of the analysis period but transitioned to mixed chemical regimes or
$NO_X$-limited conditions by the end of the analysis period. Nine areas have disappearing weekend effect trends in both
datasets and with both metrics indicating strong agreement that they are shifting to more $NO_X$-limited conditions:
Milwaukee, Houston, Phoenix, Denver, Northern Wasatch Front, Southern Wasatch Front, Las Vegas, Los Angeles –
San Bernardino County, Los Angeles – South Coast, and San Diego. The disappearing weekday effect was identified
for multiple rural and agricultural areas of California which were $NO_X$ -limited for the entire analysis period but appear
to become less influenced by local day of week emission patterns in more recent years. Finally, we discuss a variety
of reasons why there are no statistically significant trends in certain areas including complex impacts of heterogeneous
source mixes and stochastic impacts of meteorology. Overall, this assessment finds that the EQUATES modeling
simulations indicate more $NO_X$-saturated conditions than the observations but do a good job of capturing year-to-year
changes in weekend-weekday ozone patterns.

## 36    1 Introduction

Ground-level ozone ($O_3$), a key component of photochemical smog, has adverse impacts on human health and
ecosystems (U.S. Environmental Protection Agency, 2019). In the United States (US), the Clean Air Act Amendments
of 1970 instruct the Environmental Protection Agency (EPA) to set National Ambient Air Quality Standards



(NAAQS) for criteria pollutants. Since 1979, $O_3$ has served as the indicator species for the criteria pollutant of
photochemical oxidants (44 FR 8202) and since 1997, the form of the standard has been determined by the 3-year
average of the annual 4th-highest daily maximum 8-hour concentration (MDA8) (62 FR 38856). In 2015, the $O_3$
NAAQS were revised to the current level of 0.070 ppm or 70 ppb (80 FR 65291). As of 2018, 52 areas in the US had
been designated as nonattainment of the 2015 $O_3$ NAAQS (83 FR 25776; 83 FR 35136; 83 FR 52157).

$O_3$ is predominantly a secondary pollutant formed from photochemical reactions of nitrogen oxides ($NO_X$) and volatile
organic compounds (VOCs). Ground-level $O_3$ concentrations are a complex nonlinear function of the chemistry of
natural and anthropogenic precursor emissions, as well as meteorology, transport, and deposition (Seinfeld and Pandis,
2016). $O_3$ formation rates depend on the concentrations and speciation of $NO_X$ and VOCs. To reduce ambient $O_3$
concentrations, control strategies have been enacted in the US over the last 50 years to control the emissions of both
$NO_X$ and VOCs (Simon et al., 2015).

The effectiveness of different control strategies on $O_3$ production rates depends on the photochemical environment
under which ozone is formed. Ozone formation environments are typically categorized as either $NO_X$-limited or $NO_X$-
saturated, with a mixed or transitional regime between the two (Sillman, 1995, 1999; Sillman et al., 1990). In the $NO_X$-
limited regime, ambient ozone concentrations will respond more strongly to changes in $NO_X$ emissions than VOC
emissions. In contrast, in a $NO_X$-saturated (or VOC-limited) regime ozone will increase with $NO_X$ emission controls
but will decrease with VOC emissions controls. Understanding the photochemical regimes of different ozone
nonattainment areas and how they have changed over time is important for understanding the impacts of previous
control strategies and guiding future control strategies to have the maximum health benefit with the least economic
burden.

Different methods have been proposed to determine ozone formation regimes and their changes over time. One
common method used to evaluate ozone formation chemistry is through day-of-week (DOW) differences in the
concentration of ozone and its precursors. The DOW effects leverage $NO_X$ emissions differences between weekdays
and weekends (Marr and Harley, 2002a, b). In the US, onroad vehicles are a dominant source of $NO_X$ emissions (Toro
et al., 2021). Diesel vehicle traffic tends to be higher on weekdays (Monday through Friday) than on weekends
(Saturday and Sunday). This results in higher $NO_X$ emissions on weekdays than weekends (Marr and Harley, 2002a,
b). Daily varying emissions sources such as diesel vehicles are not a major source of VOC emissions. In addition,
VOC emissions in some areas are dominated by biogenic emissions that do not vary by day of week. Consequently,
VOC emissions are generally similar on weekends and weekdays in most areas. The result of DOW $NO_X$ patterns is
that ozone concentrations tend to be higher on weekends than weekdays in $NO_X$-saturated areas and lower on
weekends than weekdays in $NO_X$-limited areas (Koplitz et al., 2022). DOW differences in ozone were first reported
in the 1970s (Bruntz et al., 1974; Cleveland et al., 1974). In 2002 the DOW ozone differences in California were
explicitly tied to DOW patterns in diesel vehicle traffic (Marr and Harley, 2002a, b). Since that time, multiple studies
have used DOW ozone patterns to assess ozone chemical formation regimes in individual US cities including Los



Angeles, California (Chinkin et al., 2003; Fujita et al., 2003b; Fujita et al., 2003a; Gao, 2007; Gao and Niemeier,
2007; Warneke et al., 2013), Fresno, California (De Foy et al., 2020), Sacramento, California (Murphy et al., 2007),
Phoenix, Arizona (Atkinson-Palombo et al., 2006), Atlanta, Georgia (Blanchard and Tanenbaum, 2006), Baltimore,
Maryland (Roberts et al., 2022), and New York City, New York (Singh and Kavouras, 2022). A smaller number of
studies have assessed ozone DOW patterns across multiple US urban areas (Blanchard et al., 2008; Jaffe et al., 2022;
Koo et al., 2012; Koplitz et al., 2022; Pun et al., 2003). Additionally, ozone DOW patterns have been used as a method
for assessing chemical formation regimes outside of the US in Shanghai, China (Zhang et al., 2023), the Lesser Antilles
Archipelago (Plocoste et al., 2018), Rio de Janeiro, Brazil (Martins et al., 2015), Santiago, Chile (Rubio et al., 2011),
Andalusia, Spain (Adame et al., 2014), the Iberian Peninsula (Jiménez et al., 2005), Athens, Greece (Paschalidou and
Kassomenos, 2004) and in multiple other European cities (Pires, 2012). One complication with interpreting DOW $O_3$
patterns is that $O_3$ concentrations in urban areas are generally impacted by a mix of transport and local formation. $O_3$
transport can occur over a variety of timescales. In some locations there could be a regional $O_3$ DOW effect that might
be evident as a slightly lagged timescale depending on typical transport times from major upwind urban source areas.
Previous work has shown a substantial decrease in $NO_X$ emissions in the US over the past 20 years as a result of
national, state, and local regulations (Krotkov et al., 2016; Lamsal et al., 2015; Russell et al., 2012; Toro et al., 2021).
Concurrent with the US $NO_X$ decreases, multiple studies have found that ozone chemical formation regimes have also
changed in the US (Jin et al., 2020; Jin et al., 2017; Koplitz et al., 2022). In this paper, we focus on 51 areas in the US
which were designated in 2018 as nonattainment (https://www.epa.gov/green-book/green-book-8-hour-ozone-2015-
area-information) under the 2015 $O_3$ NAAQS (some of these areas have since been redesignated to attainment based
on clean monitoring data). We look at changes in DOW patterns in the US over 18 years from 2002 to 2019 using
both measured and modeled data to provide insights into how ozone formation chemistry has changed in the US as a
result of emissions reductions, and to assess how well modeling is able to capture the observed changes. This 18-year
dataset, which is part of EPA's Air QUAlity TimE Series Project (EQUATES), is unique in its application of consistent
emissions and modeling methodologies across the entire analysis period providing an opportunity to assess multi-year
trends.

**2 Methods**

For this assessment we use MDA8 ozone monitoring data obtained from EPA's Air Quality System (AQS)
(https://www.epa.gov/aqs) and MDA8 ozone modeling data from simulations of the Community Multiscale Air
Quality model version 5.3.2 (CMAQv5.3.2). The CMAQ model data are part of EQUATES which provides an 18-
year set of modeled meteorology, emissions, air quality and pollutant deposition spanning the years 2002 through
2019 using consistent modeling methods across years. The CMAQv5.3.2 model configuration, including input data,
boundary conditions, and science options are available from US EPA (Epa, 2021). The emissions inventories
developed for the EQUATES CMAQ modeling are described in (Foley et al., 2023).



We extract CMAQ modeling data only for days and grid-cells with monitoring data such that both datasets are paired
in time and location. Both datasets are subset to ozone monitors located within 51 of the 52 areas that were designated
in 2018 as nonattainment for the 2015 $O_3$ NAAQS (a list of areas is available in Tables S1 and S2) (83 FR 25776; 83
FR 35136; 83 FR 52157). Because this analysis focuses on May-September data, we do not include data from the
Uintah Basin nonattainment area for which violations of the NAAQS predominantly occur in winter months. Data are
analyzed for the 18-year period of the EQUATES modeling dataset.

We start by analyzing changes in ozone between weekends and weekdays pooled across all monitoring locations for
each nonattainment area for 5-year rolling periods (i.e., 14 different periods covering the 18-year timeseries). We pool
data into 5-year periods for several reasons. First, it dampens impacts of interannual meteorology that can contribute
to large year-to-year changes in ozone for a given location. Previous work has shown that differential meteorological
patterns on weekends versus weekdays impacts ozone DOW patterns in a single year and that pooling data across
multiple years can reduce this effect (Pierce et al., 2010). Second, it provides a larger sample size for calculating ozone
differences between weekends and weekdays. The use of 5-year periods does, however, limit this analysis' ability to
parse out changes in weekend-weekday differences that have occurred due to emissions changes in the most recent
individual years analyzed. For example, any changes occuring only in 2018 and/or 2019 would be dampened in the
2015-2019 pooled data.

For the purpose of quantifying differences in weekend versus weekday $O_3$ concentrations, we use Sundays to represent
weekends (WE) and Tuesdays, Wednesdays and Thursdays to represent weekdays (WD). We do not include ozone
on Monday and Saturday to minimize any carryover impacts on concentrations from the previous day and we exclude
Friday as it may exhibit somewhat different emissions patterns than the other weekdays.

We use two metrics to quantify differences in ozone between weekends and weekdays. First, we quantify mean
differences in ozone across the entire distribution of days in each season (Winter = Dec, Jan, Feb; Spring = Mar, Apr,
May, Summer = Jun, Jul, Aug, Fall = Sep, Oct, Nov, ozone season = May-Sep) using Eq. (1), where $O_{3,WE}$ represents
MDA8 $O_3$ on Sundays and $O_{3,WD}$ represents MDA8 $O_3$ on Tuesdays, Wednesdays, and Thursdays.

$$\Delta\overline{O_{3,DOW}} = \overline{O_{3,WE}} - \overline{O_{3,WD}} \tag{1}$$

In this study we mainly focus on differences during the May-Sep ozone season. The Welch's t-test (Welch, 1947) is
used to denote whether the mean WE-WD difference is statistically different from zero ($p < 0.05$). All available ozone
monitoring data and model output from all monitoring locations within each nonattainment area are included in the
calculation, providing a measure of average behavior across each area. We also examine 24-hour average modeled
formaldehyde and $NO_X$ concentrations at each of the ozone monitor locations to verify whether the model shows
expected patterns of higher $NO_X$ on weekdays than on weekends and trends in these ozone precursors. Formaldehyde
is used as an indicator of first-generation VOC reaction products for this purpose. We note that monitoring data for



VOCs and NO$_X$ are much sparser in terms of sampling frequency and spatial density than ozone measurements, so we
rely on the model alone to verify underlying day-of-week patterns in precursor compounds.

Second, similar to (Jaffe et al., 2022), we look at the percent of days with MDA8 ozone values above the NAAQS
level of 70 ppb. We calculate the percent of total weekends and weekdays in May-Sep for which MDA8 ozone
concentrations exceeded 70 ppb as shown in Eq. (2).

$\Delta O_{3,DOW,\%>70} = O_{3,WE,\%>70} - O_{3,WD,\%>70}$                                                      (2)

For this calculation, a day is characterized as exceeding the NAAQS in an area if measured and/or modeled ozone is
above 70 ppb at the location of any ozone monitor within the area. In this way we are tracking days where some
portion of the area has observed or modeled ozone above 70 ppb, but the analysis does not distinguish whether the
high ozone concentrations are localized over a small portion of the area or widespread across multiple monitoring
locations. This analysis also does not consider whether days with modeled ozone above 70 ppb occur simultaneously
with observed ozone above 70 ppb. We use the Fisher's exact test (Fisher, 1935; Mehta and Patel, 1983) to determine
whether the proportion of days above 70 ppb differs significantly (p < 0.05) between weekends and weekdays.

Next, we use the Theil-Sen estimator (Sen, 1968; Theil, 1992) to determine the multi-year trends in $\overline{\Delta O_{3,DOW}}$ and
$\Delta O_{3,DOW,\%>70}$ for each area. This nonparametric approach was chosen due to the small sample size (n=14 5-year
windows) and the fact that the Thiel-Sen estimator does not require any assumptions on the distribution of the
residuals. The Mann-Kendall test (Kendall, 1975; Mann, 1945) is used to determine the statistical significance of the
derived trends in WE-WD O$_3$ differences. For each derived trend, we also document the 95% confidence interval.
Finally, investigation of relationships between WE-WD O$_3$ and meteorological parameters used the meteorological
dataset developed by and described in (Wells et al., 2021).

**3 Results**

**3.1 NO$_X$ and formaldehyde day-of-week patterns**

For all but one of the 51 areas, the model shows clear patterns of higher NO$_X$ concentrations on weekdays than
weekends and relatively constant formaldehyde concentrations across May-Sep days for the entire 2002-2019 analysis
period. This is consistent with the underlying assumption in the ozone day-of-week analyses discussed above. Here
we describe examples of the NO$_X$ and formaldehyde day of week patterns using the data for Denver, CO and Los
Angeles, CA to show typical patterns in large urban areas and Butte County, CA to show a typical pattern in a more
rural area in Figures 1, 2, and 3, respectively. The modeled WE-WD differences in NO$_X$ concentrations are more
pronounced in large urban areas such as Los Angeles and Denver than in rural or agricultural areas such as Butte





County. The only area that does not demonstrate higher modeled $NO_X$ concentrations on weekdays than weekends is
Door County, WI (Figure S-1). Higher $NO_X$ emissions on weekdays are typically associated with commuting patterns
and greater vehicular activity from commercial truck traffic. The nonattainment portion of Door County, which was
fully redesignated to attainment in 2022 (87 FR 25410), is located at the tip of a peninsula on Lake Michigan and a
rural recreation and tourist destination (i.e., likely to see more weekend activity). Consequently, the area does not
follow typical weekday-weekend emission patterns and therefore modeled $NO_X$ concentration patterns are unlike those
of other areas. While the model does not predict substantial day-of-week formaldehyde differences in most areas,
there are small modeled weekday formaldehyde enhancements in some areas such as Chicago (Figure S-2).
Theil-Sen trends show that differences in WE versus WD $NO_X$ have diminished significantly over time in most areas
(e.g. Figures 1, 2 and 3). The WE versus WD differences in formaldehyde are also diminishing over time but to a
much lesser extent. As total emissions have decreased, absolute concentrations of $NO_X$ have also decreased. Figures
S-5 and S-6 show that the WE versus WD $NO_X$ trends remain significant whether tracking absolute or normalized
$NO_X$ differences in Denver and Los Angeles, which is consistent with WE-WD $NO_X$ trends seen in all but ten of the
nonattainment areas. In nine areas (Houston, TX; Las Vegas, NV; Muskegon, MI; New York, NY; Phoenix, AZ; San
Diego, CA; St. Louis, MO-IL; Tuolumne County, CA; and Yuma, AZ) while absolute WE-WD $NO_X$ differences have
diminished significantly there is no significant trend in relative WE-WD differences. In Mariposa County, CA neither
absolute nor relative WE-WD $NO_X$ differences have significant trends from 2002-2019.

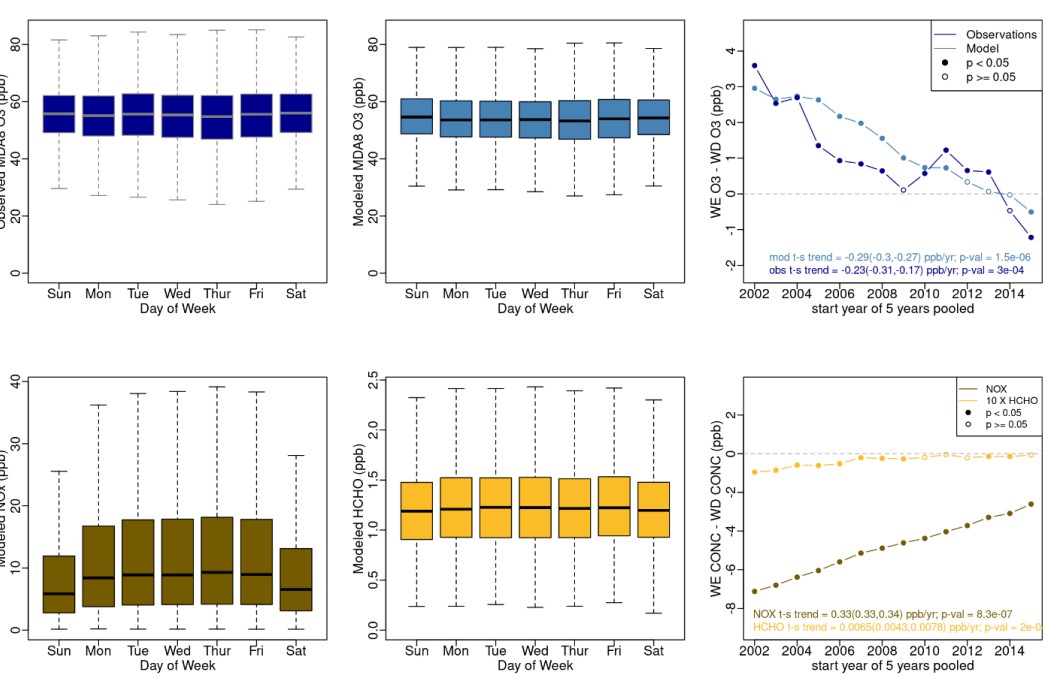






**Figure 1. Denver area 2002-2019 May-Sep: observed (top left) and modeled (top center) MDA8 ozone distribution by day**
**of week; modeled NOX (bottom left) and modeled formaldehyde (bottom center) distribution by day of week; observed and**
**modeled trends in $\Delta\overline{O_{3,DOW}}$ (top right); modeled trends in WE-WD NOX and formaldehyde differences (bottom right). The**
**distributions by day of the week are for the entire 18 years with each box representing the 25th to 75th percentile for that**
**day of the week across all 18 years, the whiskers representing the 1.5 times the interquartile range, and the bold line inside**
**the box representing the median. WE-WD differences (top and bottom right) are based on 5-year rolling periods.**

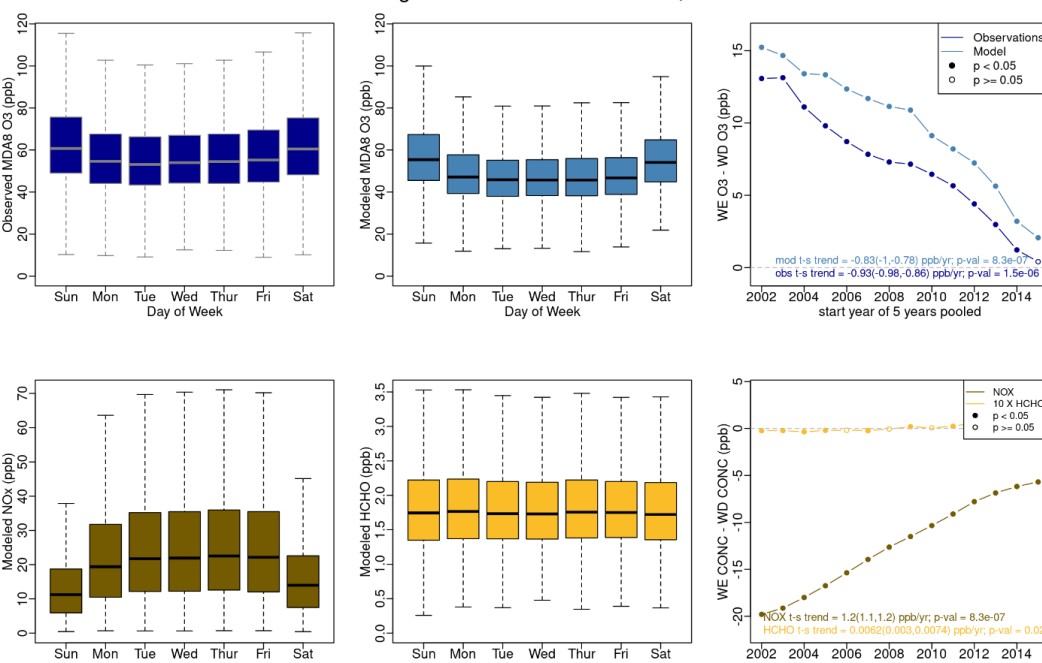


**Figure 2. Los Angeles area 2002-2019 May-Sep: observed (top left) and modeled (top center) MDA8 ozone distribution by**
**day of week; modeled NOX (bottom left) and modeled formaldehyde (bottom center) distribution by day of week; observed**
**and modeled trends in $\Delta\overline{O_{3,DOW}}$ (top right); modeled trends in WE-WD NOX and formaldehyde differences (bottom right).**
**The distributions by day of the week are for the entire 18 years with each box representing the 25th to 75th percentile for**
**that day of the week across all 18 years, the whiskers representing the 1.5 times the interquartile range, and the bold line**
**inside the box representing the median. WE-WD differences (top and bottom right) are based on 5-year rolling periods.**

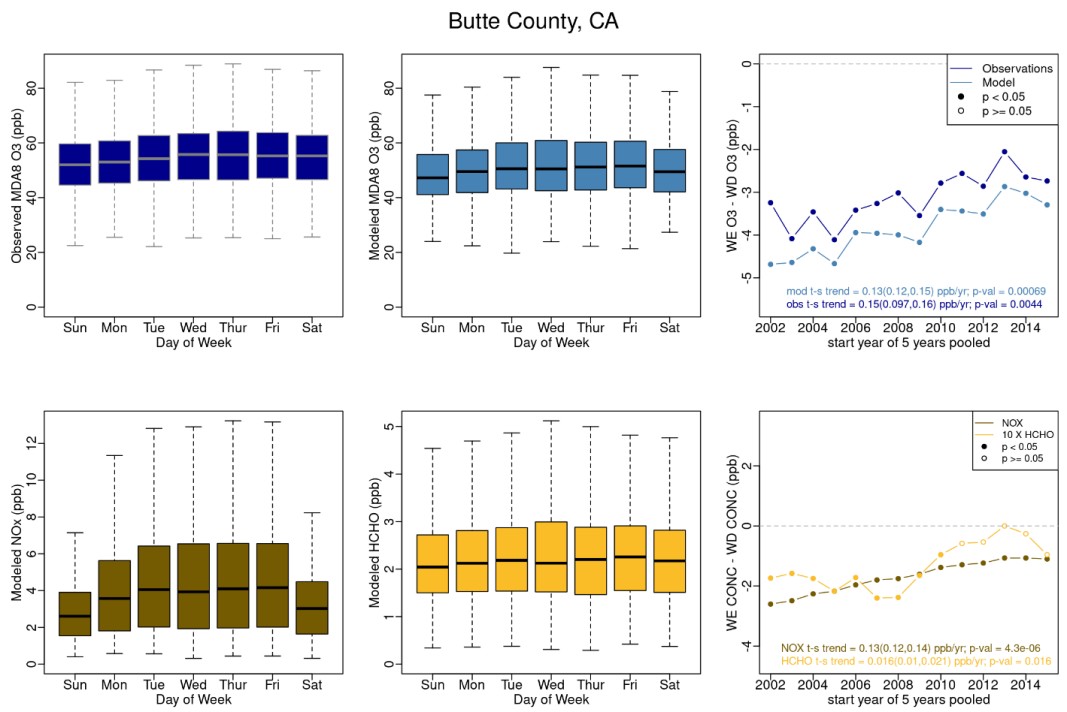

**Figure 3. Butte County, CA area 2002-2019 May-Sep: observed (top left) and modeled (top center) MDA8 ozone distribution by day of week; modeled NO$_X$ (bottom left) and modeled formaldehyde (bottom center) distribution by day of week; observed and modeled trends in $\overline{\Delta O_{3,DOW}}$ (top right); modeled trends in WE-WD NO$_X$ and formaldehyde differences (bottom right). The distributions by day of the week are for the entire 18 years with each box representing the 25$^{th}$ to 75$^{th}$ percentile for that day of the week across all 18 years, the whiskers representing the 1.5 times the interquartile range, and the bold line inside the box representing the median. WE-WD differences (top and bottom right) are based on 5-year rolling periods.**

## 3.2 Trend types of ozone day-of-week patterns

Within any 5-year window, NO$_X$-saturated areas display a "weekend effect" meaning that ozone concentrations were statistically higher on weekends than on weekdays and NO$_X$-limited areas display a "weekday effect" meaning that ozone concentrations were statistically higher on weekdays than on weekends. We categorize the trends in ozone DOW patterns into 3 discrete categories: 1) disappearing weekend effect (i.e. areas that went from NO$_X$-saturated to NO$_X$-limited), 2) disappearing weekday effect (i.e. areas that went from NO$_X$-limited to approaching zero in terms of DOW differences), and 3) areas with no significant change over the 18-year time period. Disappearing weekend effect areas are characterized by a negative Thiel-Sen slope (e.g. Denver and Los Angeles in Figures 1 and 2 respectively). Disappearing weekday effect areas are characterized by a positive Thiel-Sen slope (e.g. Butte County in Figure 3). Areas with no trend are characterized by a lack of significance, as determined by the Mann-Kendall test. Trend types for all 51 areas based on observed and modeled datasets are shown in Figure 4 and 5.



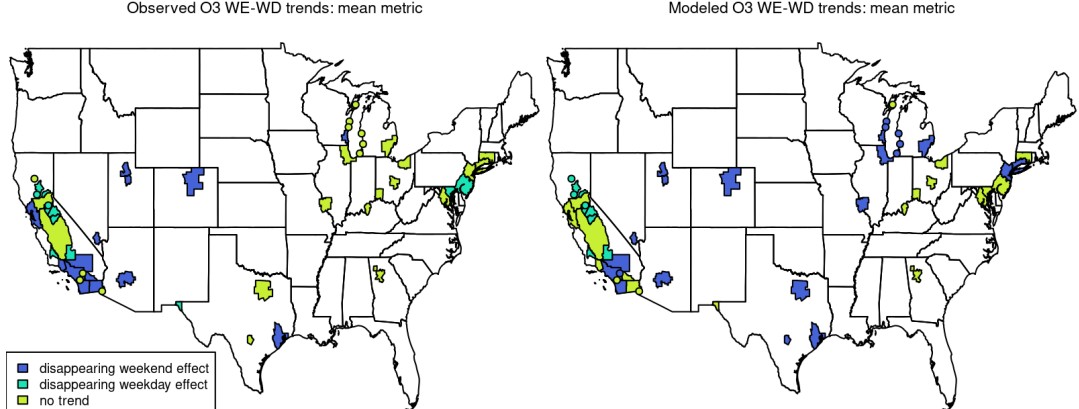


**Figure 4. Map of ozone nonattainment areas color coded by trends in mean ozone day of week differences ($\Delta\overline{O_{3,DOW}}$) using**
**observed data (left) and modeled data (right) over an 18-year period from 2002-2019. Ozone nonattainment areas less than**
**3000 km$^2$ in area are shown as dots on the map for visibility.**

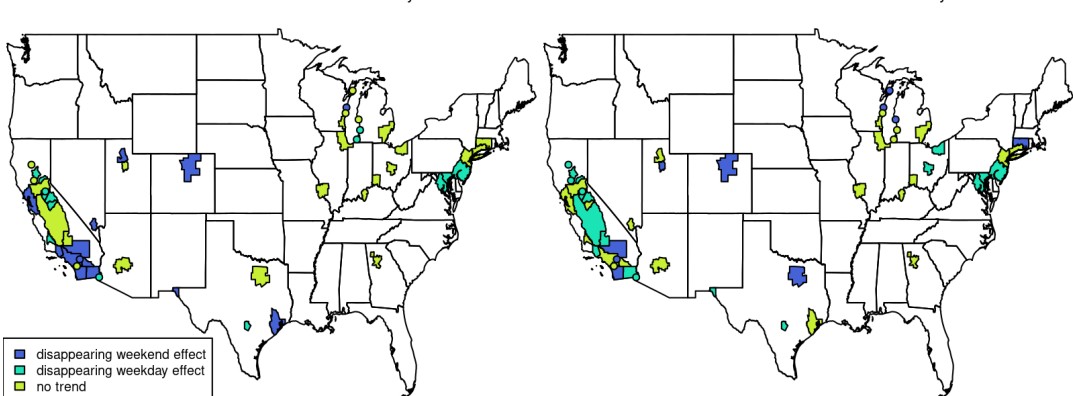


**Figure 5. Map of ozone nonattainment areas color coded by trends in ozone day of week differences based on the percentage**
**of days >70 ppb MDA8 ($\Delta O_{3,DOW,\%>70}$) using observed data (left) and modeled data (right) over an 18-year period from**
**2002-2019. Ozone nonattainment areas less than 3000 km$^2$ in area are shown as dots on the map for visibility.**

**3.2.1 "Disappearing weekend effect" case studies**

The disappearing weekend effect trend is typical of areas that initially had strongly positive ozone WE-WD differences
(i.e., mean MDA8 ozone is higher on weekends than on weekdays), suggesting NO$_X$-saturated conditions, at the
beginning of the analysis period. These areas typically transition into significant negative WE-WD MDA8 O$_3$
differences by the most recent 5-year window, suggesting a shift to NO$_X$-limited conditions by the end of the analysis
period. Of the 51 nonattainment areas analyzed, 15 exhibit this type of trend based on observed data and 23 based on
modeled data for $\Delta\overline{O_{3,DOW}}$. Two areas that exhibit this trend for $\Delta\overline{O_{3,DOW}}$ are Denver and Los Angeles shown in



Figures 1 and 2 respectively. In Denver, the modeled and observed $\Delta\overline{O_{3,DOW}}$ are statistically significant and in the
range of +3 to +4 ppb at the beginning of the analysis period. Both the model and observed data have statistically
significant decreasing Thiel-Sen slopes for $\Delta\overline{O_{3,DOW}}$, -0.29 ppb/yr and -0.23 ppb/yr for Denver and Los Angeles
respectively. In the most recent 2015-2019 5-year window, both modeled and observed $\Delta\overline{O_{3,DOW}}$ are negative and
statistically different from zero, suggesting a shift to $NO_X$-limited conditions. While the results shown in Figure 1
represent aggregated measured ozone data across all Denver nonattainment area monitors, Figure 6 shows behavior
at three specific monitors in Denver with monitoring records covering the majority of the analysis period. All three
sites were located to the south and southwest of the Denver urban area. The Welch monitor is located closer to the
Denver urban area in proximity to two major highways. While the monitored and modeled negative Thiel-Sen slopes
for $\Delta\overline{O_{3,DOW}}$ holds at all 3 sites, there are differences in the magnitude of the slopes and the sign of $\Delta\overline{O_{3,DOW}}$ across
sites. For instance, the Welch and Highland Reservoir sites both have statistically significant positive $\Delta\overline{O_{3,DOW}}$ at the
beginning of the analysis period suggesting both sites were $NO_X$-saturated in the early 2000s. While the Chatfield site
had positive $\Delta\overline{O_{3,DOW}}$ at the beginning of the analysis period, the differences were not statistically different from zero,
suggesting that this location may have already been transitioning to $NO_X$-limited conditions in the early-to-mid 2000s.
The model predicts that all three sites have non-significant negative $\Delta\overline{O_{3,DOW}}$ at the end of the analysis period while
observations show the negative $\Delta\overline{O_{3,DOW}}$ to be statistically significant at Chatfield and Highland Reservoir. This
suggests that the model may understate the $NO_X$-limited conditions in recent years at these locations. Los Angeles
provides another example of an area where both the model and the observations had strongly positive $\Delta\overline{O_{3,DOW}}$ at the
beginning of the analysis period and disappearing weekend effect trends (Figure 2). Similar to Denver, site to site
differences in the magnitude of $\Delta\overline{O_{3,DOW}}$ are evident in Los Angeles (Figure S-7) but the disappearing weekend effect
trend is fairly consistent across sites. Similar types of trends in Chicago and Houston are shown in supplemental
figures S-2 and S-3.

In general, similar disappearing weekend effect trends in $\Delta O_{3,DOW,\%>70}$ are evident, however this metric appears to be
noisier perhaps because it is capturing the frequency of extreme events which have a more stochastic nature than mean
ozone differences. Specifically, since there are a low number of exceedance days for most nonattainment areas in any
given year, a metric based on the percentage of those days falling on a Sunday versus a Tuesday, Wednesday or
Thursday will be inherently more noisy than a metric based on mean values. Figures 7 and 8 show $\Delta O_{3,DOW,\%>70}$
Thiel-Sen trends for Denver and Los Angeles. In both cases, the model underpredicts both the percentage of days with
MDA8 $O_3$ > 70 ppb and the Thiel-Sen slope. Additional examples of results for $\Delta O_{3,DOW,\%>70}$ are provided for
Chicago, Houston and New York City in Figure S-9, S-10 and S-11 respectively.




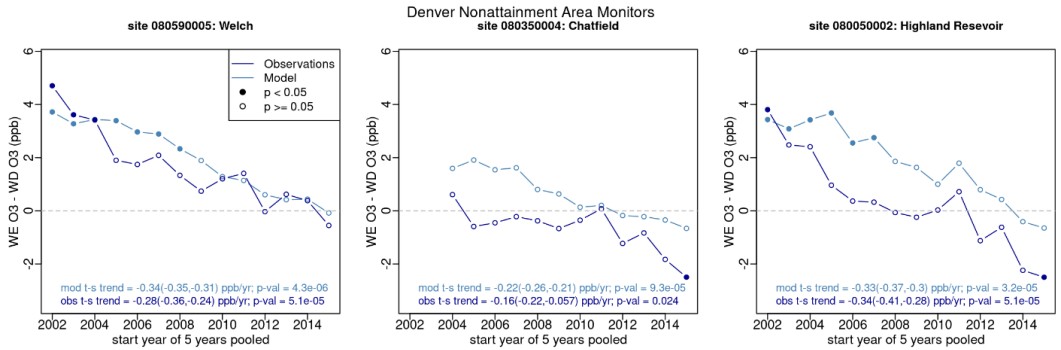

**Figure 6.** Observed and modeled May-Sep trends in mean ozone day of week differences ($\overline{\Delta O_{3,DOW}}$) at three Denver area monitoring locations for 2002-2019 plotted as 5-year rolling periods.

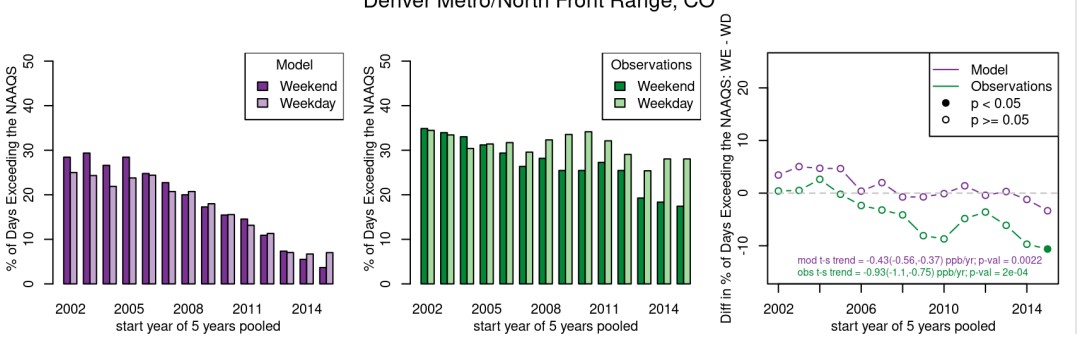

**Figure 7.** Modeled (left) and observed (center) percent of days with MDA8 ozone exceeding 70 ppb at any monitor within the Denver nonattainment area during May-Sep on weekends and weekdays for 5-year rolling periods between 2002-2019; Observed and modeled trends in May-Sep $\Delta O_{3,DOW,\%>70}$ at Denver area monitors for 5-year rolling periods between 2002-2019 (right).

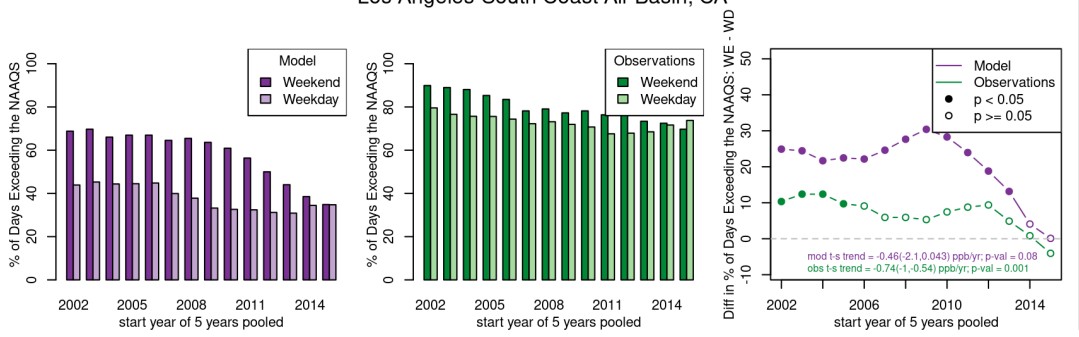

**Figure 8.** Modeled (left) and observed (center) percent of days with MDA8 ozone exceeding 70 ppb at any monitor within the Los Angeles nonattainment area during May-Sep on weekends and weekdays for 5-year rolling periods between 2002-2019; Observed and modeled trends in May-Sep $\Delta O_{3,DOW,\%>70}$ at Los Angeles area monitors for 5-year rolling periods between 2002-2019 (right).



### 3.2.2 "Disappearing weekday effect" case study

The disappearing weekday effect trend type in $\Delta\overline{O_{3,DOW}}$ is evident in 12 out of the 51 nonattainment areas using observed data and 11 out of the 51 nonattainment areas using modeled data (Figure 4). This trend type is characterized by negative $\Delta\overline{O_{3,DOW}}$ values (i.e., weekday ozone higher than weekend ozone) throughout the analysis period indicating NO$_X$-limited conditions trending upwards toward zero which appears primarily in rural/agricultural areas in California. The Butte County nonattainment area in California is one example of an area exhibiting this type of day-of-week trend pattern as is evident using both $\Delta\overline{O_{3,DOW}}$ and $\Delta O_{3,DOW,\%>70}$ (Figures 3 and 9 respectively). The disappearing weekday effect could indicate that sources without day-of-week activity patterns are becoming more dominant contributors to local NO$_X$ emissions. In that case, the day-of-week patterns for ambient NO$_X$ concentrations are becoming less pronounced which would result in reductions in day-of-week ozone patterns. An alternate explanation is that local NO$_X$ emissions in general have decreased substantially enough that local ozone formation has become less important in such areas and a larger fraction of total ozone is being transported from upwind sources. In that case, the origin of the transported ozone could be a mixture of multiple source areas that are at varying distances upwind which could lead to a loss in the day-of-week ozone signal. More analysis would be needed to investigate this idea with respect to nonattainment areas of interest.

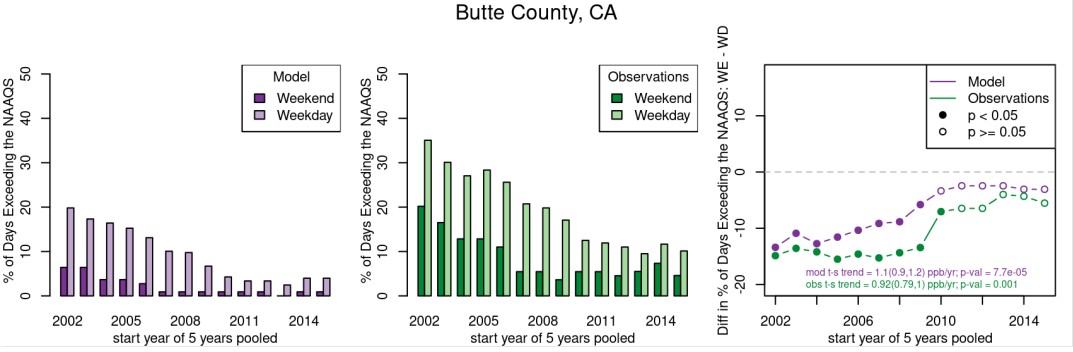

**Figure 9. Modeled (left) and observed (center) percent of days with MDA8 ozone exceeding 70 ppb at any monitor within the Butte County, CA nonattainment area during May-Sep on weekends and weekdays for 5-year rolling periods between 2002-2019; Observed and modeled trends in May-Sep $\Delta O_{3,DOW,\%>70}$ at Butte County, CA area monitors for 5-year rolling periods between 2002-2019 (right).**

### 3.2.3 "No trend" case studies

Out of the 51 nonattainment areas analyzed, 25 do not have a statistically significant $\Delta\overline{O_{3,DOW}}$ trend based on a p-value cut-off of 0.05 using observed data and 18 do not have a statistically significant trend using modeled data. The reason for the lack of trends may vary by area. Plots for several areas are provided in the supplemental information. Figures S-4, S-8 and S-11 provide the analysis for New York City which shows no trend for the $\Delta\overline{O_{3,DOW}}$ using observations but a statistically significant disappearing weekend effect trend for this metric using modeled data. Neither the model nor the observations show a significant trend in $\Delta O_{3,DOW,\%>70}$. One possible explanation for the lack of trends in New



York is the complex nature of the emissions sources and the meteorology impacting ozone formation in this area.
Figure S-8 shows $\Delta\overline{O_{3,DOW}}$ trends at three monitors in the New York City nonattainment area occuring in very
different locations. The Bronx IS 52 monitor, which is located in an urbanized part of the nonattainment area, shows
significant disappearing weekend effect in both modeled and observed $\Delta\overline{O_{3,DOW}}$. In contrast the Long Island –
Riverhead monitor and the Bridgeport CT monitor are both located in portions of the nonattainment area that are
typically downwind of the urban core on high ozone days and are impacted by complex meteorology associated with
the land-water interface near the Long Island sound. The modeled and observed data do not show significant
$\Delta\overline{O_{3,DOW}}$ trends at the Long Island site and only the model shows disappearing weekend effects trends at the CT site.
Due to the complex nature of this large urban area, some sites may not show trends at all and trends at other sites may
be masked when aggregating data across a large number of sites.

Several nonattainment areas appear to have negative slopes in $\Delta\overline{O_{3,DOW}}$ at the beginning of the analysis period and
positive slopes at the end of the analysis period resulting in no overall trend taken over the entire period. Cincinnati,
OH-KY exemplifies this pattern and on closer inspection the patterns appear to mirror annual changes in WE-WD
patterns in multiple meteorological parameters (Figure S-12). For Cincinnati the correlation coefficients between WE-
WD MDA8 $O_3$ differences and WE-WD meteorological parameter differences were 0.77, -0.83, 0.79, 0.89, -0.94, and
-0.73 for daily maximum temperature, daily average relative humidity, daily maximum planetary boundary layer
height, solar radiation, percent cloud cover and 24-hour transport direction respectively. Other areas exhibiting this
behavior are all located in relatively close proximity to Cincinnati, including Louisville, KY-IN and St. Louis, MO-
IL and to a lesser extent Columbus, OH and Atlanta, GA. These findings suggest that for these areas even five-year
processing blocks may not be sufficient to remove the effects of spurious weekly meteorological variations on ozone.
Figure S-13 shows that the correlation between WE-WD differences in seven meteorological variables and observed
$\Delta\overline{O_{3,DOW}}$ do not appear to be a driving factor in significant $\Delta\overline{O_{3,DOW}}$ trends in other areas but it is possible that some
additional areas which do not have statistically significant trends in $\Delta\overline{O_{3,DOW}}$ may also be impacted by meteorological
variations.

**367   3.3 Comparison of modeled and observed trends in ozone day-of-week patterns**

The modeled and observed trends in WE-WD differences for each of the 51 nonattainment areas are provided in
supplemental tables S1 ($\Delta\overline{O_{3,DOW}}$) and S2 ($\Delta O_{3,DOW,\%>70}$). Figure 10 provides a comparison of modeled to observed
WE-WD differences across the 51 nonattainment areas at the beginning of the analysis period (2002-2006) and at the
end of the analysis period (2015-2019). Each point represents the WE-WD MDA8 ozone difference for a single
nonattainment area, with the left-hand panel showing $\Delta\overline{O_{3,DOW}}$ and the right-hand panel showing $\Delta O_{3,DOW,\%>70}$. Data
points falling in the upper right quadrant of each panel represent areas for which both the observations and the modeled
DOW patterns suggest $NO_X$-saturated conditions. Data points in the lower left quadrant of each panel represent areas
for which both the observations and the model DOW patterns suggest $NO_X$-limited conditions. In the earlier 2002-
2006 time period, there are a large number of areas falling in both the upper right and lower left quadrants for both





metrics. In the 2015-2019 time period, almost all areas are located in the lower left quadrant for both metrics
suggesting that most US nonattainment areas have transitioned into NO$_X$-limited conditions. The correlation of
modeled and observed WE-WD differences is quite high (r = 0.94 and 0.82 for $\Delta\overline{O_{3,DOW}}$ in the earliest and most recent
time periods, respectively, and r = 0.7 and 0.62 for $\Delta O_{3,DOW,\%>70}$ in the earliest and most recent time periods,
respectively). For both metrics, the majority of points fall above the 1:1 line indicating that, in general, the model
overestimated the degree of NO$_X$-saturated conditions and underestimated the degree of NO$_X$-limited conditions.

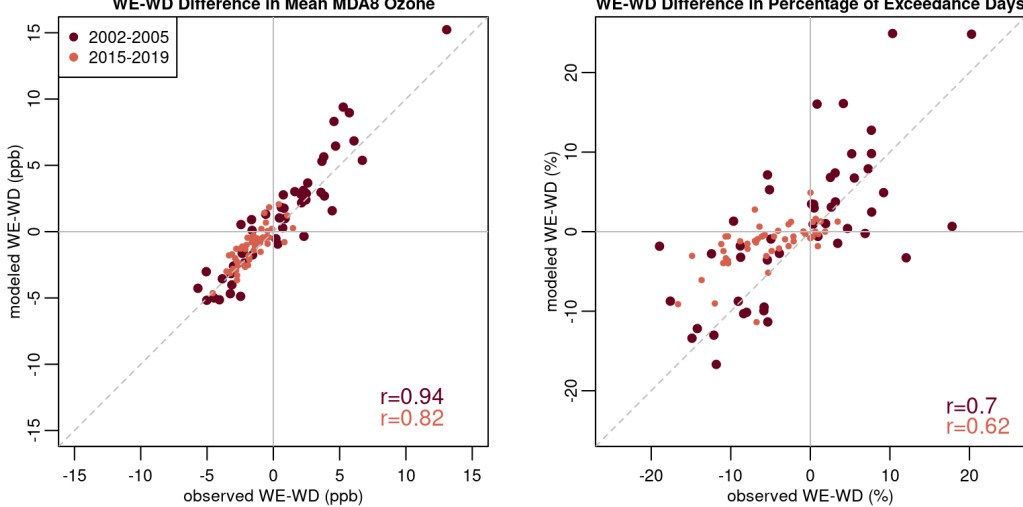


**Figure 10. Comparison of modeled and observed WE-WD MDA8 O$_3$ differences for $\Delta\overline{O_{3,DOW}}$ (left panel) and $\Delta O_{3,DOW,\%>70}$**
**(right panel). Differences shown for the 2002-2006 time period and for the 2015-2019 time period. Each dot represents a**
**different nonattainment area.**

Maps in Figures 4 and 5 show the locations of areas predicted to have disappearing weekend effect trends, disappearing
weekday effect trends and no trends for $\Delta\overline{O_{3,DOW}}$ and $\Delta O_{3,DOW,\%>70}$ respectively. The maps show general consistency
among which areas are predicted to have each trend type between observations and the model, although some areas
predicted to have significant trends with one dataset or with one metric do not have significant trends with the other
dataset or metric. Nine areas are predicted to have disappearing weekend effect trends in both datasets and with both
metrics indicating strong agreement that they are shifting to more NO$_X$-limited conditions: Milwaukee, WI; Houston,
TX; Phoenix, AZ; Denver, CO; Northern Wasatch Front, UT; Southern Wasatch Front, UT; Las Vegas, NV; Los
Angeles – San Bernardino County, CA; Los Angeles – South Coast, CA; and San Diego, CA.

Figure 11 compares modeled and observed Thiel-Sen slopes in WE-WD MDA8 O$_3$ differences across all areas. Each
point represents a single nonattainment area color-coded by median $\Delta\overline{O_{3,DOW}}$ or $\Delta O_{3,DOW,\%>70}$. The correlation of
modeled versus observed Thiel-Sen slopes using $\Delta\overline{O_{3,DOW}}$ is stronger (r = 0.8) than the correlation using $\Delta O_{3,DOW,\%>70}$
(r = 0.47). While the model does not always correctly predict the Thiel-Sen slope, the data falls close to the 1:1 line
for the $\Delta\overline{O_{3,DOW}}$ suggesting that the model does not systematically over or under predict the trends in WE-WD





differences from 2002-2019. The trend types described above for $\Delta\overline{O_{3,DOW}}$ metric are visible in the left-panel of Figure
11. Most $NO_X$-saturated areas (yellow and brown symbols) and some $NO_X$-limited areas (blue symbols) have negative
Thiel-Sen slopes towards $NO_X$-limited conditions similar to those described above for Denver and Los Angeles
(shown as the dark brown symbol at the bottom-left of the plot). Areas with positive Thiel-Sen slopes tend to be the
most $NO_X$-limited areas (darker blue symbols) and represent the disappearing weekday trends demonstrated by Butte
County. The model is not as accurate at predicting $\Delta O_{3,DOW,\%>70}$ Thiel-Sen slopes as $\Delta\overline{O_{3,DOW}}$ Thiel-Sen slopes, as
evidenced by the increased scatter in the right-hand panel of Figure 11 compared to the left-hand panel. Some areas
have few exceedances of the NAAQS in the later years of the trends period and this small sample size could explain
the difference between the monitored and modeled slopes, given that the model predicted fewer exceedance days than
were observed in many areas.

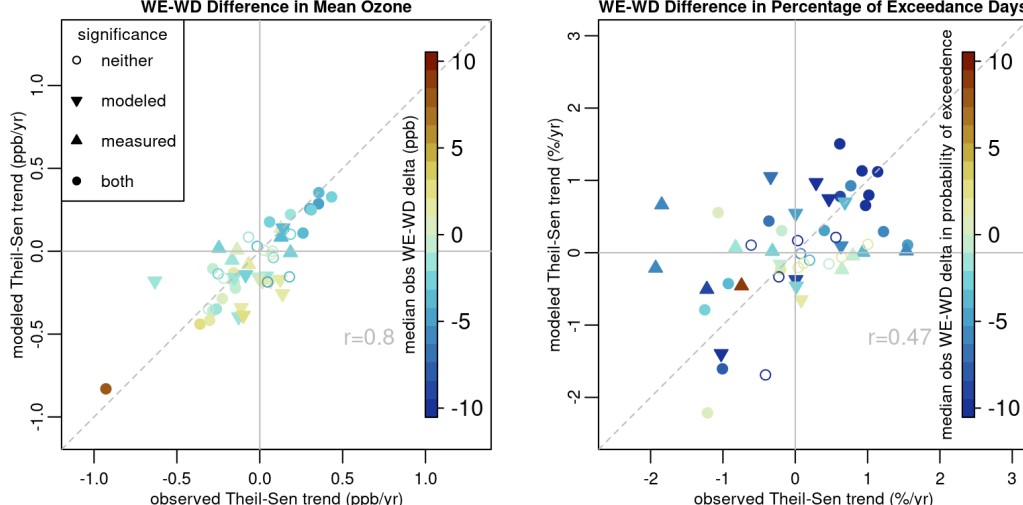


**Figure 11. Comparison of modeled and observed Thiel-Sen slopes in May-Sep WE-WD MDA8 $O_3$ differences across all nonattainment areas for $\Delta\overline{O_{3,DOW}}$ (left panel) and $\Delta O_{3,DOW,\%>70}$ (right panel). Whether or not the trend is significant with observed and/or modeled data is indicated by the shape of the symbol. Median WE-WD differences across all years are indicated by the color scale with positive differences ($NO_X$-saturated areas) shown in shades of yellow and brown and negative differences ($NO_X$-limited areas) shown in shades of blue. Note that the brown symbol on both figures represents the Los Angeles nonattainment area.**


Figure 12 shows the comparison of $\Delta\overline{O_{3,DOW}}$ Thiel-Sen slopes by season. The summer plot looks similar to the May-
September plot shown in Figure 11. Winter, spring, and fall data show median $\Delta\overline{O_{3,DOW}}$ near zero or greater than zero
in most nonattainment areas suggesting transitional or $NO_X$-saturated conditions in these seasons. Both observations
and model predictions suggest $\Delta\overline{O_{3,DOW}}$ negative Thiel-Sen slopes in these seasons suggesting that nonattainment
areas in the US may be transitioning towards $NO_X$-limited conditions even outside of the summer ozone season.

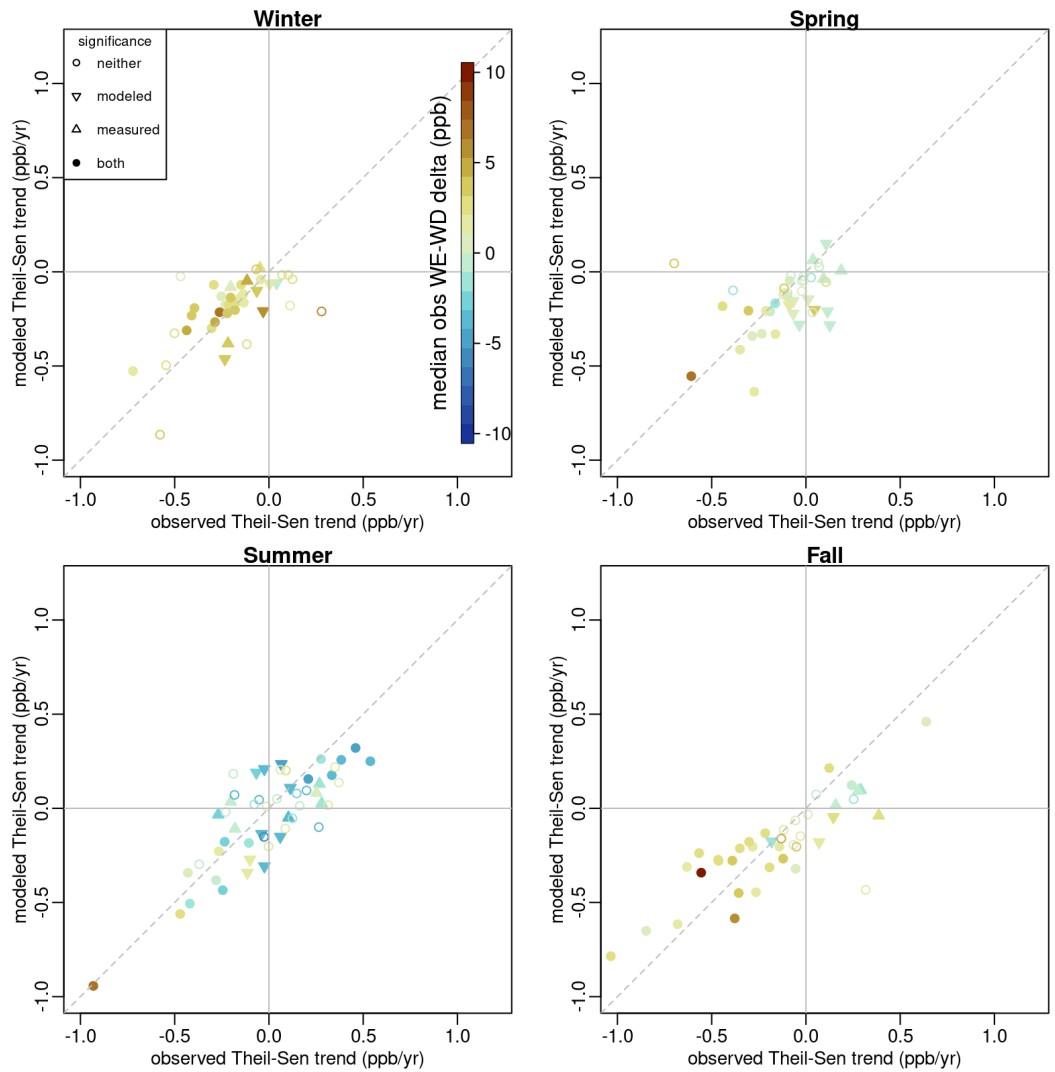

**Figure 12. Comparison of modeled and observed** $\Delta \overline{O_{3,DOW}}$ **Theil-Sen slopes across all nonattainment areas in winter (top left), spring (top right), summer (bottom left) and fall (bottom right). Whether or not the trend is significant with observed and/or modeled data is indicated by the shape of the symbol. Median WE-WD differences across all years are indicated by the color scale with positive differences (NO$_X$-saturated areas) shown in shades of yellow and brown and negative differences (NO$_X$-limited areas) shown in shades of blue. Note that year-round ozone monitoring is not required in some parts of the US and therefore monitoring data may not be available outside the May-September period in some areas.**

**4 Conclusions**

While this assessment has provided insight into the ozone formation regimes across high-ozone locations in the US, some key questions remain about the important drivers for year-to-year changes in DOW ozone patterns and which of those drivers are well captured by the EQUATES dataset. First, while NO$_X$ and VOC emissions have been steadily decreasing across most areas of the US, exceptions to that pattern include increasing wildfire emissions especially in





the Western US and increasing emissions from oil and gas activities near US nonattainment areas in Texas, Colorado,
New Mexico and Utah. Future work could focus on areas impacted by these two emissions sources to assess both the
impact of these increasing emissions on ozone formation regimes and the ability of the EQUATES dataset to capture
those impacts. Second, this assessment predominantly focused on ozone values across the May-Sep ozone season,
however, past work has identified some seasonally varying ozone biases within the CMAQ model (Appel et al., 2021).
Specifically, EQUATES has a tendency to underpredict ozone during the spring and overpredict ozone later in the
summer (Figures S-14 and S-15). Given that ozone formation tends to be more $NO_X$-saturated in the springtime than
in the summer (Jin et al., 2020; Jin et al., 2017), a more in-depth assessment would be needed to fully characterize the
extent that differences in observed and modeled WE-WD ozone differences are impacted by this seasonally varying
model performance. Third, we assessed DOW ozone patterns across multiple complex urban areas that encompassed
spatially heterogeneous emissions sources and meteorology. For some of these areas (e.g. Los Angeles, CA and
Denver, CO) the sign of the Thiel-Sen slopes in WE-WD ozone appeared consistent across monitoring locations while
in others (e.g. New York City, NY) different monitoring locations across the area appeared to show different types of
trends. Further local scale investigation into each of these areas would be necessary to fully characterize the nuances
of DOW and year-to-year variations in emission and meteorology that obscure the ozone DOW trends in some areas
but not others when aggregating across monitor locations in those areas. Finally, an intriguing trend in ozone DOW
patterns was identified in multiple rural and agricultural areas of California. Recent literature has suggested that soil
NO emissions, which are unlikely to have a DOW emissions pattern, are an important $NO_X$ emissions source in
agricultural locations of California (Almaraz et al., 2018; Zhu et al., 2023). Could the ozone DOW trends observed in
these areas be reflective of the increasing relative importance of $NO_X$ sources other than mobile sources in those
locations? More assessment is needed to definitively determine whether the trend in a decreasing weekday effect is a
reliable indicator of areas that are becoming more dominated by local $NO_X$ sources that do not vary by DOW, more
dominated by transported ozone, or some other factor. It is important to note that transported ozone may come from
nearby regional sources or from longer range sources provided the transport times are sufficient to mask any DOW
patterns that would be evident in the source region.

In this analysis we found that trends in ozone formation chemistry may not always be clearly shown by trends in DOW
patterns which are impacted by a complex set of local factors including meteorology, the mix of local emissions
sources and monitor locations in relationship to land-water interfaces. Lack of trends appear more often using observed
data than modeled data (Figures 4 and 5) meaning that, while the model accurately captures Thiel-Sen slopes for
$\Delta\overline{O_{3,DOW}}$ and $\Delta O_{3,DOW,\%>70}$ (Figure 11), p-values below 0.05 are less common using observational data. This suggests
that there may be some stochastic processes making observed year-to-year WE-WD ozone differences noisy which
are not fully captured by the model. Even with these limitations, this analysis has shown that DOW patterns in ambient
$NO_X$ concentrations persist in US urban areas but have become less prominent over the 18-year period analyzed. These
DOW $NO_X$ differences have resulted in distinctive DOW ozone patterns in many of the nonattainment areas assessed.
The EQUATES modeling simulations appear to show larger and more positive WE-WD ozone differences than
observational data suggesting that ozone formation in this modeling dataset is less $NO_X$-limited than in the



observations. Despite this discrepancy, the EQUATES dataset captures year-to-year changes in WE-WD ozone
patterns as demonstrated by high correlation of the Thiel-Sen slopes for WE-WD ozone differences. Both the WE-
WD ozone trends and agreement between the modeled and observation datasets are more apparent when assessing
summertime mean MDA8 ozone than when analyzing extreme values using the percentage of exceedance days metric.
Assessing frequencies or magnitudes of extreme values is challenging using a dataset with a limited number of
weekend and weekday days due to the stochastic and infrequent nature of high ozone events in many areas.

While there are multiple types of measurements and modeling assessments that can be applied to characterize local
ozone formation regimes, many of these require specialized measurements or datasets that are not readily available in
all areas. In contrast, assessing DOW ozone patterns requires only routine daily ozone measurements that are widely
available across urban areas in the US and in other countries. Consequently, this type of assessment is a useful tool
and may be applied in many areas using routine measurements. In locations with long-term measurements, DOW
patterns offer a method to look at trends in ozone formation chemistry over time. While DOW patterns in ozone are
especially useful given the wide availability of data required for this type of assessment, we anticipate that in the near
future additional datasets for assessing ozone chemical formation regimes will become more widely available.
Specifically, $O_3$, $NO_2$ and HCHO data from the recently launched TEMPO satellite may provide the ability to better
understand the relationships between WE-WD ozone patterns and precursor concentrations.
**Author contributions**
All authors contributed to conceptualization of the project.  HS, CH, KF, BW, and WA contributed to data curation.
HS conducted formal analysis.  HS, CH, AW, KF, BW, BH, and SK contributed to developing the methodology.
HS and BW developed software for performing the analysis.  HS, CH, AW, JL, NP, BW, and GT contributed to
validation. HS, BW, and BH helped visualize the data. All authors contributed to the writing and editing of the
manuscript.
**Competing interests**
The authors declare that they have no conflict of interest.

*Disclaimer:* The views expressed in this manuscript are those of the authors and do not necessarily reflect the views
or policies of the U.S. Environmental Protection Agency.
**Acknowledgements**
The authors would like to acknowledge Chris Nolte and Golam Sarwar for helpful comments on this manuscript.

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

Theil, H.: A Rank-Invariant Method of Linear and Polynomial Regression Analysis, in: Henri Theil's
Contributions to Economics and Econometrics: Econometric Theory and Methodology, edited by: Raj, B.,
and Koerts, J., Springer Netherlands, Dordrecht, 345-381, 10.1007/978-94-011-2546-8_20, 1992.



Toro, C., Foley, K., Simon, H., Henderson, B., Baker, K. R., Eyth, A., Timin, B., Appel, W., Luecken, D.,
Beardsley, M., Sonntag, D., Possiel, N., and Roberts, S.: Evaluation of 15 years of modeled atmospheric
oxidized nitrogen compounds across the contiguous United States, Elementa-Science of the
Anthropocene, 9, 10.1525/elementa.2020.00158, 2021.
U.S. Environmental Protection Agency: Integrated Science Assessment (ISA) for Particulate Matter (Final
report, Dec 2019). U.S. Environmental Protection Agency, Washington, DC, EPA/600/R-19/188, 2019.
Warneke, C., de Gouw, J. A., Edwards, P. M., Holloway, J. S., Gilman, J. B., Kuster, W. C., Graus, M., Atlas,
E., Blake, D., Gentner, D. R., Goldstein, A. H., Harley, R. A., Alvarez, S., Rappenglueck, B., Trainer, M., and
Parrish, D. D.: Photochemical aging of volatile organic compounds in the Los Angeles basin: Weekday-
weekend effect, 118, 5018-5028, https://doi.org/10.1002/jgrd.50423, 2013.
Welch, B. L.: THE GENERALIZATION OF 'STUDENT'S' PROBLEM WHEN SEVERAL DIFFERENT POPULATION
VARLANCES ARE INVOLVED, Biometrika, 34, 28-35, 10.1093/biomet/34.1-2.28, 1947.
Wells, B., Dolwick, P., Eder, B., Evangelista, M., Foley, K., Mannshardt, E., Misenis, C., and Weishampel,
A.: Improved estimation of trends in U.S. ozone concentrations adjusted for interannual variability in
meteorological conditions, Atmospheric Environment, 248, 118234,
https://doi.org/10.1016/j.atmosenv.2021.118234, 2021.
Zhang, G., Sun, Y., Xu, W., Wu, L., Duan, Y., Liang, L., and Li, Y.: Identifying the O3 chemical regime
inferred from the weekly pattern of atmospheric O3, CO, NOx, and PM10: Five-year observations at a
center urban site in Shanghai, China, Science of The Total Environment, 888, 164079,
https://doi.org/10.1016/j.scitotenv.2023.164079, 2023.
Zhu, Q., Place, B., Pfannerstill, E. Y., Tong, S., Zhang, H., Wang, J., Nussbaumer, C. M., Wooldridge, P.,
Schulze, B. C., Arata, C., Bucholtz, A., Seinfeld, J. H., Goldstein, A. H., and Cohen, R. C.: Direct
observations of NOx emissions over the San Joaquin Valley using airborne flux measurements during
RECAP-CA 2021 field campaign, Atmos. Chem. Phys. Discuss., 2023, 1-21, 10.5194/acp-2023-3, 2023.













