# Peer review of "Revisiting Day-of-Week Ozone Patterns in an Era of Evolving U.S. Air Quality"

_EGUsphere, 2023_

## Author Comment (AC1)

**Comment:** In this analysis the authors evaluate day of week patterns in average O3 and O3 exceedance days for the 2002-2019 timeframe at ~50 different high O3 sites in the U.S. The authors identify several different patterns including disappearing weekend-weekday differences and others. The analysis mixes observations and model results, which I find to be somewhat problematic. In addition I have some statistical concerns that would need to be corrected or clarified before this could be published.

**Response:** See responses to specific comments below

**Comment:** The authors mix observed patterns with model patterns in a way that I believe is misleading. I think its incumbent on the authors to first clearly document what the observations show. Then we can ask how well the model reproduces the observations and what we can learn from the model where it is consistent with the observations, or if not consistent, then why. Certainly there are plenty of NOx observations that could have been used for this work (see Jaffe et al 2020), so I am not sure what is gained by showing and using only the modeled NOx. But NOx is not that big of a concern. Its formaldehyde that I find much more problematic. For formaldehyde, we have much poorer understanding of emissions and chemistry, both of which are essential to understanding the concentrations. Without any evaluation of the modeled formaldehyde, these results should be removed. In other places the authors quote both modeled trends and observed trends and appear to put equal weight on these. That is incorrect, in my opinion, for the reasons stated above.

**Response:** Thank you for this comment. We will address the comment in regards to $NO_x$ and Formaldehyde analyses separate from ozone analysis.

We start by looking at $NO_x$ and formaldehyde day-of-week patterns to confirm whether the ozone precursor concentrations follow the expected decreases on weekends that would be necessary to drive an ozone day-of-week effect. We explain in methods that due to sparsity of monitoring data we chose to use modeled values for this analysis. However, based on the reviewer comment we have added a new analysis to the supplemental information (Figure S-1 through S-26) that compares modeled and observed $NO_X$ DOW patterns and trends in locations where measurements are available. We note that this does not include all nonattainment areas from this analysis and within nonattainment areas it does not include all locations with ozone monitors. However, the new figures in the supplemental information show that the model does a reasonable job of representing the $NO_X$ DOW patterns and trends across these areas. We were unable to add a similar analysis for formaldehyde because until very recently most formaldehyde measurements occurred at NATTS and urban air toxics monitoring networks (https://www.epa.gov/amtic/air-toxics-monitoring-national-program-reports) which have 1-in-6 and 1-in-12 day sampling schedules which does not provide sufficient data frequency for DOW analysis. In recent years PAMS has required HCHO at 1-in-3 day interval during June-Aug but most sites did not start meeting these requirements until between 2017 and2019. We have clarified throughout section 3.1 that results represent modeling rather than observed data and have added the following text to the beginning of section 3.1 to clarify these points:

*We first look at modeled NOx and formaldehyde day-of-week patterns to better understand how daily changes in precursor emissions impact modeled day-of-week ozone patterns. We chose to focus on modeled data here because of the ubiquitous spatial and temporal coverage provided in the model for these pollutants allowing us to evaluate these pollutants on the same days and at the same locations as the ozone monitors. We note that some observed NOx data can also be used for this purpose, although the available NOx data are not available for all nonattainment areas and are not available at the*

*locations of all ozone monitors even within nonattainment areas with some NOx monitoring data. A comparison of monitored and observed trends in NOx day-of-week differences provided in Figures S-1 through S-26 show that the model does reasonably well at capturing the patterns in the limited observational dataset that is available. Due to the sparsity of formaldehyde measurements, both spatially and temporally (formaldehyde is commonly measured at a 1-in-6 day or 1-in-12 day frequency), a similar comparison cannot be made for modeled and measured formaldehyde. However, with more recent requirements for formaldehyde measurements at Photochemical Assessment Monitoring Stations (PAMS) locations starting in the 2017-2019 time-period future assessments may have additional measured formaldehyde data that could be used for this purpose.*

For the ozone analysis, we think it is appropriate to present the observational results side-by-side with modeled results. We are always clear on which results come from the monitoring data versus the model and do not believe this is misleading in any way. Rather we think that the current structure allows for better flow of the manuscript and highlights which aspects of the observed trends the model captures and which it does not.

**Comment:** I believe the authors may over-state some of the statistical significance due to auto-correlation. This could be true for both the t-tests for individual years and the trends, which use 5-year running means.

**Response:** Thank you for this comment. We agree that the auto-correlation is an issue for the Theil-Sen trends using the 5-year rolling time periods. This is a challenge because the single-year values are subject to meteorological and other random effects which are minimized by using the five year rolling windows. We do believe that the trends shown by the Theil-Sen method show real changes over time as they are consistent with patterns shown in Figure 10 which compares WE-WD differences in 2002-2005 vs 2015-2019 which do not have any overlapping data points. However, we think it is important to acknowledge this weakness and are thus de-emphasizing any language implying "statistical significance" of trends and rather are simply reporting P-Value ranges which still provides some indication of which areas have stronger trends. We note that reporting P-values while de-emphasizing "statistical significance" is also consistent with the recommendations of Reviewer 1 and the approach taken in TOAR. We have added the following language to the methods section to acknowledge this issue: "*Because we use a 5-year rolling window for each area, the individual data points in the trends analysis are correlated. While this should not systematically bias the calculated slopes, it will lead to lower P-values and narrower 95% confidence intervals than would be calculated if the data points were uncorrelated. However, the P-value is still informative to characterize which areas have the strongest trends. Therefore, while we do report P-values we do not rely on a strict threshold for determining statistical significance.*"

We developed an additional analysis that demonstrates auto-correlation is not an important issue for the individual year t-tests. The t-tests for individual years compare T/W/Th weekday ozone to Sunday weekday ozone. The most important autocorrelation issues would occur on the transition days (i.e. Mon, Fri, Sat) and would degrade the statistical significance, which is why those days are excluded from the WE-WD difference calculation. To show that the T/W/Th correlation does not impact our results, we performed a second analysis in which we used the average of O3 on T/W/Th of each week rather than individual T/W/Th O3 values to compare against Sunday O3. By averaging T/W/Th (instead of pooling), we create weekly data points days that would be more independent. The figure below shows the

individual year t-test results from the original analysis compared to the analysis using a T/W/Th average and demonstrates that results do not meaningfully change. We therefore have chosen to retain the original method which used all T/W/Th O3 values. It is important to note that we cannot perfectly eliminate autocorrelation (e.g., monthly), but longer term autocorrelation would degrade statistical significance because it would manifest as a residual component of both weekday and weekend.

[Figure]

**Comment:** Finally I note that this has a lot of overlap with our earlier analysis (Jaffe et al 2020). We used data for 1995-2020. This analysis uses data for 2002-2019. There are some modest differences, but overall the results are quite similar. I think its essential that the authors clearly describe what is new and/or whether these results are consistent with the earlier analysis. One area that is different is use of probability of exceedance vs mean concentration. The authors seem to want to discount any differences as being due to random variability, but I am not sure that is true. One focuses on the highest days and the other approach focuses on all days in the O3 Do these days have the same VOC-NOx sensitivity?

**Response:** Thank you for this response. We think there are multiple important novel aspects of our work compared to the analysis presented by Jaffe et al (2022). First, in our work we endeavor to examine areas individually and highlight nuances in behavior driven by local factors while Jaffe et al (2022) mostly provided national-level results. We agree that the "transitioning chemical regime" trend (formally called the "disappearing weekend effect") is broadly consistent with the national results

reported by Jaffe et al (2022) and have added a statement acknowledging this work in the first paragraph of section 3.2.1. The "disappearing weekday effect" trend that we report in rural/agricultural areas of California was not identified in Jaffe at al (2022) and we are not aware of it being reported anywhere in the literature. Jaffe et al focused their analysis on areas with $NO_x$ measurements and many of the areas displaying this trend type do not have $NO_x$ monitors and were not included in the Jaffe et al analysis. Additionally, we examine local features that have led to no trends in DOW patterns in some nonattainment areas. We believe that the local analysis of O3 DOW patterns in individual areas is a key, unique aspect of our analysis that has allowed us to better understand the varied local factors leading to different trends in different areas. A further unique factor of our work is that we evaluate trends using both a mean metric and the percentage of exceedance days metrics to show that trends are broadly consistent across not only high ozone days but also when looking across the entire ozone season. Finally, Jaffe et al focused on observed data while the inclusion of both modeled and observed data allows us to evaluate the skill of the CMAQ model at capturing these patterns of changing ozone chemical formation regimes which has important policy-relevant implications since many regulators use CMAQ modeling as part of planning for ozone control strategies. Demonstrating the skill of this model builds confidence in our ability to use it as a tool for this purpose. The model additionally allows us to better characterize drivers of observed trends since we are able to probe the model in locations where there are no measurements available. For instance we can confirm that the trends in DOW O3 patterns in the model are occuring coincident with expected trends in modeled NOx and formldehyde ozone precursors.

**Comment:** Abstract: It is important in abstract to describe the scope: All US O3, all US urban areas or all US non-attainment areas. What years? How many regions considered? In addition, I am unclear what it means if you have a "disappearing weekday" effect. The information here is contained in the relative O3 and NOx behavior between weekday and weekend. So the terms "disappearing weekday" is confusing.

**Response:** The abstract does include the scope of the analysis "across US nonattainment areas" (we have added "51" before "US nonattainment areas" to clarify the large number of areas analyzed) and also include the years of analysis and the metrics. We do not break out our results by region but rather report results from individual areas so regions are not mentioned in the abstract. We agree that the term "disappearing weekday effect" was confusing and have revised the name of this trend type to "transitioning chemical regime" to better convey that negative slopes in WE-WD ozone represent areas that are transitioning from VOC-limited conditions to NOx-limited conditions.

**Comment:** Line 25-26: "both datasets" ?

**Response:** We have updated the language to clarify this is using both observed and modeled data.

**Comment:** Line 27: The abstract uses area names that are consistent (I think) with EPA designations, but are often rather non-intuitive. For example, Los Angeles – San Bernardino County vs Los Angeles – South Coast. The San Bernardino monitors are in Riverside CBSA, so aren't these two locations essentially same region. It would be more interesting to include a site closer to downtown LA like Azusa, where we might expect a different pattern.

**Response:** We think that using the official area names and delineations consistent with EPA nonattainment designations provides a consistent framework with which to distinguish areas. Within

each nonattainment area we include data from all available monitoring site locations. The Azusa monitoring data is included in the Los Angeles South Coast nonattainment area.

**Comment:** Line 33: It is not clear what model evaluation for this work. As near as I can tell, nothing was shown about the models ability to capture year-year variations. The model does seem to capture the trend in weekend-weekday differences.

**Response:** We include multiple comparisons of the model's prediction of ozone (and now NOx) DOW patterns and trends with observed patterns and trends. These comparisons are provided in every figure within the paper. The term "year-to-year variations" captures both the trends in DOW differences and the areas in the Ohio River Valley region of the country without trends but with multi-year meteorology-driven patterns (i.e. Figures S-28 and S-29). We additionally include plots showing model evaluation for ozone across years, seasons, and regions of the US in Figures S-40 and S-41.

**Comment:** 132-135: While I understand why you excluded 3 out of 7 days, does this change the results?

**Response:** Due to concerns with autocorrelation between days that this reviewer has brought up, we did not evaluate how including M, F, and Sat would impact the results.

**Comment:** 144: Not clear how t-tests were done. I think you took every weekday and weekend day in one year and compared the means and treated each day as an independent observation. If this is right, then I don't think autocorrelation was taken into account. In any case please clarify how the t-tests were performed.

**Response:** Yes, your understanding of how the t-tests were performed is correct. As shown in our response above, we conducted a sensitivity analysis using the mean of T/W/Th for each week and found results did not change. We have added the following sentence to the methods clarifying the t-test calculation: "Within each nonattainment area, the t-test calculation compared the means of every weekday and every weekend day in a 5-year window, treating each day as an independent observation."

**Comment:** 169: Given the 5 year running means, these values will have sig autocorrelation. Was this taken into account in the results?

**Response:** As explained above, due to the autocorrelation issue and comments from Reviewer #1 we have decided to de-emphasize the use of strict P-value thresholds for determining statistical significance and have added the following discussion of limitations to the methods section: "Because we use a 5-year rolling window for each area, the individual data points in the trends analysis are correlated. While this should not systematically bias the calculated slopes, it will lead to lower P-values and narrower 95% confidence intervals than would be calculated if the data points were uncorrelated. However, the P-value is still informative to characterize which areas have the strongest trends. Therefore, while we do report P-values we do not rely on a strict threshold for determining statistical significance."

**Comment:** 173-174: Unclear meaning.

**Response:** We have added the following sentence to more fully explain the meteorology analysis: "Meteorological parameters were similarly compared across weekends and weekdays, matching times and locations of the ozone analysis and using the same statistical methods for comparison."

**Comment:** 193-194: Unclear meaning.

**Response:** We have modified this sentence for clarity as follows: "While the model does not predict substantial day-of-week formaldehyde differences in most areas, there are small modeled formaldehyde enhancements on weekdays compared to weekends in some areas such as Chicago (Figure S-28)"

**Comment:** Figure 1: Please clarify meaning of P values in top right plot. I think these are for each individual year, correct? Given that the NOx and CH2O plots are for all years, not sure what is the value in showing these. There are major differences (for NOx) between the early and later part of the data record.

**Response:** The p-values come from the t-test results for each 5-year window to show whether the WE-WD differences are statistically different from zero. We have added a sentence clarifying this to the caption of each figure.

**Comment:** 234: As noted above, the terminology "disappearing weekend effect" is very misleading. Its really about the difference between weekend and weekday values.

**Response:** Thank you for this comment. We have updated the term used for this trend type to "transitioning chemical regime" to better convey that negative slopes in WE-WD ozone represent areas that are transitioning from VOC-limited conditions to NOx-limited conditions.

**Comment:** 258-259: So how do we interpret these model obs differences? You may say the random variations impact the obs more than the model, but aren't these variations important?

**Response:** Figures 10 and 11 show that the model generally captures the DOW O3 trends across most areas although the model does not perfectly simulate the patterns in every case.

**Comment:** 283: I don't think the probability approach is inherently noisier, especially when averaged over several years as you have done. I think this is an interesting spot to do a deeper dive.

**Response:** We initially based the statement on the observation that there were more areas falling into the "no trend" category for the probability approach. However, we have taken a closer look at the results and agree that the probability approach results do not look inherently noisier in the observations although the model has less skill at replicating this behavior. We have deleted this sentence.

**Comment:** Conclusions: As noted above it would be good to understand what is new here. Please add some discussion to clarify, perhaps focusing on the differences between the prob of an exceedance approach and mean O3 approach.

**Response:** Within the results section, we now note where our results are in broad agreement with Jaffe et al (2022). We also note in the section discussing the "disappearing weekday effect" that we are not aware of this being reported anywhere in the literature.

**Comment:** Finally, I note that the regression information in the right plots of figures 1-9 (not 5) is almost impossible to read.

**Response:** We have removed this text from the plots. Readers can find now find this information in Tables S-1 and S-2.

---

## Author Comment (AC2)

**Comment:** This is a very good study, written by experts at EPA and it will be a welcome addition to the literature. I find the research question to be important and the conclusions are clearly supported by the analysis. I don't have any concerns regarding the analysis or conclusions, my only recommendation is that the authors update their terminology regarding the reporting of statistical findings, to be more consistent with the Tropospheric Ozone Assessment Report and current thinking regarding the limitations of the expression "statistically significant", as described below.

**Response:** Thank you for this comment.

**Comment:** Regarding the use of the Theil-Sen/Mann-Kendall method for calculating trends, the authors state that they chose this method because of the small sample sizes and because it does not require assumptions about the distribution of the residuals. Another reason that is often given for the choice of this method is that it is resistant to outliers. The problem is that in order to remove the impact from outliers, this method automatically ignores up to 29% of the data points in a sample (see Section 2 of Chang et al., 2021). This would be fine if the analyst believed that the outliers are due to instrument errors, but there is no reason to throw out data if they are believed to be reliable. In your case there is no reason to believe that your samples contain erroneous data points that should be ignored. For this reason, the Tropospheric Ozone Assessment Report has abandoned the Theil-Sen/Mann-Kendall method that was used in the first phase of TOAR (2014-2019). A further problem with the Theil-Sen method is that it produces unrealistically narrow 95% confidence intervals. This is shown in Figure 1 of the TOAR-II Recommendations for Statistical Analyses (available at https://igacproject.org/activities/TOAR/TOAR-II). Figure 1 compares the trend and 95% confidence interval calculated by 10 different methods for the ozone time series at Mace Head, Ireland. The Theil-Sen method has the narrowest 95% confidence interval by far, and the reason is that this method ignores 29% of the data; by throwing out all of the extreme values the sample has very little variability and therefore a straight line can be fit through the remaining data within a very narrow range. The second phase of TOAR-II is now recommending the use of quantile regression, as described in the TOAR-II Recommendations for Statistical Analyses. Quantile regression was used to good effect in the very nice paper by co-author B. Wells (Wells et al., 2021), and it could easily be applied to your current analysis.

**Response:** Thank you for this comment. As stated in the manuscript, we believe that Theil-Sen/Mann-Kendall methods are appropriate due to the relatively small sample sizes in contrast to the large ozone datasets used in the TOAR analysis. However, based on the reviewer's suggestions we repeated our analysis using quantile regression. We found that regression slopes between Theil-Sen and quantile regression were nearly identical. When comparing P-values we found that they did differ between methods but did not find any systematic bias towards higher or lower P-values with one method versus the other. Importantly, most regressions that had significant slopes in our original analysis using either a P-Value cutoff of 0.05 or 0.1 still had significant slopes when using quantile regression. Similarly, most areas that had insignificant slopes in our original analysis also had insignificant slopes with the quantile regression method. Based on these results, we have opted not to update the regression methods used in this manuscript. Full results from this comparison are provided below.

**Mean Metric: slopes**

[Figure]

[Figure]

**Mean Metric: p-values**

[Figure]

[Figure]

**Mean Metric: p-values**

|  | p-value | N- model | N -obs |
|---|---|---|---|
| Number of areas with regressions | N/A | 52 | 52 |
| Number of areas with significant slopes in both regression types | 0.05 | 28 | 22 |
|  | 0.1 | 30 | 25 |
| Number of areas with insignificant slopes in both regression types | 0.05 | 17 | 24 |
|  | 0.1 | 12 | 19 |
| Number of areas with significant MK slope and insignificant QR slope | 0.05 | 4 | 4 |
|  | 0.1 | 6 | 3 |
| Number of areas with insignificant MK slope and significant QR slope | 0.05 | 2 | 1 |
|  | 0.1 | 3 | 4 |

**% exceedance days metric: slopes**

[Figure]

% exceedance days metric: Observations

y = 0.98x + 0.0055, r = 0.96

[Figure]

% exceedance days metric: Model

y = 1x + 0.0084, r = 0.99

**% exceedance days metric : p-values**

[Figure]

% exceedance days metric: Observations

$y = 1.2x + -0.0047, r = 0.88$

p-value using MannKendall

p-value using quantile regressions

[Figure]

% exceedance days metric: Model

$y = 0.8x + 0.0055, r = 0.83$

p-value using MannKendall

p-value using quantile regressions

**% exceedance days metric: p-values**

| | p-value | N- model | N -obs |
|---|---|---|---|
| Number of areas with regressions | N/A | 52 | 52 |
| Number of areas with significant slopes in both regression types | 0.05 | 26 | 25 |
| | 0.1 | 29 | 29 |
| Number of areas with insignificant slopes in both regression types | 0.05 | 20 | 22 |
| | 0.1 | 16 | 19 |
| Number of areas with significant MK slope and insignificant QR slope | 0.05 | 3 | 4 |
| | 0.1 | 5 | 4 |
| Number of areas with insignificant MK slope and significant QR slope | 0.05 | 3 | 1 |
| | 0.1 | 2 | 0 |

**Comment:** Throughout the paper the authors use the expression "statistically significant", however this expression is now recognized as being problematic and it should be abandoned and replaced by the more useful method of reporting all trends (with uncertainty) and all p-values, followed by a discussion of the trends and the author's opinion regarding their confidence in the trend values. This advice comes from a highly influential paper by Wasserstein et al. (2019), published in the journal, The American Statistician, that has already been cited over 1300 times (according to Web of Science). This advice was adopted by the first phase of TOAR (Tarasick et al., 2019) and will also be used by TOAR-II. Some other recent papers on ozone trends that have taken this advice are: Chang et al., 2020; Cooper et al., 2020; Gaudel et al., 2020; Chang et al., 2022; Wang et al., 2022; Mousavinezhad et al., 2023. Because these papers report all trend values, uncertainties, and all p-values, and also discuss the trend results, there is no confusion regarding the findings, and one does not even notice that the term "statistically significant" is not used at all.

The authors describe a trend as "no trend" when the p-value is greater than 0.05. There are two problems with this approach:

1) as described above the expression "statistically significant" which is tied to the p-value of 0.05 should be abandoned. Just because a trend has a p-value of 0.06, it does not mean that there is absolutely no trend, it just means that there is a gray area and the trend is not as robust as one that has a p-value of 0.02. Chang et al. (2017) provide a nice demonstration of the useful information that can be gleaned from a trend with a p-value greater than 0.05 (see their Figure 13). They calculated a regional ozone trend for the eastern USA using all available ozone monitors (in summer the trend was strongly negative for the period 2000-2014). They then conducted an exercise to see what would happen to the regional trend if they threw out all time series with a p-value less than 0.05. The result was almost the same because the time series with p-values greater than 0.05 still reflected the overall regional decrease of ozone.

2) The authors are using the Theil-Sen method to calculate trends and p-values. As described above the 95% confidence intervals are unrealistically narrow using this method, and therefore the p-values are also too low. This means that too many sites are classified as having a real trend, according to the 0.05 p-value threshold. If the authors use another method for calculating trends (like quantile regression) the p-values will increase and they would then have to classify more sites as having "no trend". Given the gray area around p-values, and given that trends with p-values greater than 0.05 can still be reliable, there is no justification for dichotomizing ozone time series as "trend" or "no trend" based on a p-value.

When I look at the maps in Figure 4 and 5 I am left wondering about the non-attainment regions labeled as "no trend". Is there really no trend here, i.e. a flat line, or is there still a decrease, but it just doesn't reach the arbitrary threshold of p<0.05? A good example is Tuscan Buttes. Table S-1 shows the observed and modelled trend is the same (0.14) but because the model has a p-value of 0.02 this trend is considered to be real, while the observations have a p-value of 0.06 and are classified as "no trend". The TOAR papers report all trends and all p-values and the trend values in their map plots are colored according to p-value (Fleming et al., 2018). This allow the reader to see if a trend is still notable (e.g. a p-value between 0.05 and 0.10) or if there really and truly is no trend (e.g. a p-value > 0.33). It would be very helpful to the reader if the authors can color their maps according to p-value, in a manner similar to TOAR.

**Response:** Thank you for this comment.  We have revised the maps in Figures 4 and 5 to show the P-value ranges from the TOAR assessment: P <= 0.05, 0.05 < P <= 0.1, 0.1 < P <= 0.33, and P > 0.33.  We have also revised the timeseries symbols in figures 1, 2, 3, 6, 7, 8 and 9 to use symbols representing these four P-value ranges.  We have attempted to remove the term "statistically significant" wherever possible and instead just report the P-value ranges.  We now define the "no trend" areas using a threshold of P > 0.33.  We have also removed the symbols indicating statistical significance from Figures 11 and 12.

---

## Author Comment (AC3)

The submitted manuscript undertakes an interesting analysis – an examination of day-of-week patterns in ozone concentrations across many US regions. The goal is to illuminate possible changes in regional photochemical environments due to transitions between NOX versus VOC sensitive ozone formation. The authors present a great deal of analysis of both observational data and model results, and reach important conclusions, e.g., "Nine (large US urban) areas have disappearing weekend effect … indicating .. that they are shifting to more NOX-limited conditions: …." However, there are shortcomings in the authors' analysis; below we discuss two of these that compromise the robustness of the author's conclusions if they cannot be adequately addressed.

**Comment:** First, a significant fractional "disappearance" in observed day-of-week ozone patterns must be expected simply due to the decrease in anthropogenic ozone formation driven by effective precursor emission controls implemented over the past decades. Parrish et al. (2016) show that between 1980 and 2015 in the Los Angeles urban area (i.e., California's South Coast Air Basin) the temporal dependence of the anthropogenic ozone contribution to the observed distribution of MDA8 ozone concentrations is well-described by an exponential decay with an e-folding time of 22 years. Decreases in anthropogenic ozone formation of similar magnitudes have been documented in other US regions (Parrish et al., 2022, and references therein). In the submitted manuscript, the authors consider 18 years (2002 to 2019) during which this decrease would have amounted to a factor of 2.3. Since the authors use an absolute measure of the day-of-week ozone pattern (i.e., mean difference in ozone in ppb between weekends and weekdays, their Equation 1), that measure would be expected to decrease by that same factor, or by 56% over their selected analysis period, even in the absence of any change in photochemical environment. Unless the authors can demonstrate that the disappearance of day-of-week ozone pattern is significantly greater (or lesser) than 56% of the anthropogenic contribution, those disappearances cannot be taken as evidence for a change in the photochemical environment of the ambient atmosphere.

**Response:** Thank you for this comment. We believe that the original "disappearing weekend effect" terminology used in our manuscript was confusing as it implied that the weekend effect became less pronounced over time when, in fact, for most areas the "weekend effect" (i.e. higher ozone on weekends than weekdays) turned into a "weekday effect" (i.e. higher ozone on weekdays than weekends) (for example see Figure 1 for Denver). Given that the sign of the WE-WD ozone difference changed for most areas, we don't think that adding the additional comparison to 56% suggested above would provide a meaningful change to the results or conclusions. We have however changed the terminology throughout the paper to "transitioning chemical regime" trend to help clarify the trend pattern we are describing.

**Comment:** Second, the details of the trend analysis require further discussion. The authors analyze 18 years (2002 to 2019) of observed MDA8 ozone concentrations. However, rather than analyze the 18 individual years, the authors choose to base their analysis on 5-year rolling periods (i.e., 14 different periods covering the 18-year time series). Importantly, there is a great deal of autocorrelation between the 14 different 5-year rolling means. In fact, the 18 years of observations gives fewer than 4 (i.e., 3.6) independent 5-year means. Attempts to derive trends with reliable confidence limits from such a limited number of independent data is quite uncertain. It must be fully appreciated that the use of 5-year rolling means does not improve the confidence limits of derived trends over those of trends derived from the individual years. In particular, use of the Mann-Kendall test to determine the statistical significance of the derived trends in WE-WD O3 differences will give overly optimistic results if the 14 different 5-year

rolling periods are considered to be independent data. The rolling means can give plots that apparently illustrate well-defined trends (e.g., upper right graphs in Figures 1-3), but these illustrations are misleading if the autocorrelation of the 5-year means are not fully considered. A complete discussion of the trend analysis and derivation of confidence limits based upon less than 4 independent observational data points is required; if the autocorrelation of the 5-year rolling means has not been adequately considered, then revisions are required.

**Response:** Thank you for this comment.  We agree that the auto-correlation is an issue for the Theil-Sen trends using the 5-year rolling time periods. This is a challenge because the single-year values are subject to meteorological and other random effects which are minimized by using the five-year rolling windows.  We do believe that the trends shown by the Theil-Sen method show real changes over time as they are consistent with patterns shown in Figure 10 which compares WE-WD differences in 2002-2005 vs 2015-2019 which do not have any overlapping data points.  However, we think it is important to acknowledge this weakness and are thus de-emphasizing any language implying "statistical significance" of trends and rather are simply reporting P-Value ranges which still provide some indication of which areas have stronger trends.  We note that reporting P-values while de-emphasizing "statistical significance" is also consistent with the recommendations of other reviewers and the approach taken in TOAR. We have added the following language to the methods section to acknowledge this issue: "Because we use a 5-year rolling window for each area, the individual data points in the trends analysis are correlated.  While this should not systematically bias the calculated slopes, it will lead to lower P-values and narrower 95% confidence intervals than would be calculated if the data points were uncorrelated.  However, the P-value is still informative to characterize which areas have the strongest trends.  Therefore, while we do report P-values we do not rely on a strict threshold for determining statistical significance."